# A brief note on the $G_2$ Affleck–Kennedy–Lieb–Tasaki chain

Hosho Katsura[1,2,3] and Dirk Schuricht[4]

1 Department of Physics, Graduate School of Science, The University of Tokyo, 7-3-1, Hongo, Bunkyo-ku, Tokyo, 113-0033, Japan
2 Institute for Physics of Intelligence, The University of Tokyo, 7-3-1, Hongo, Bunkyo-ku, Tokyo, 113-0033, Japan
3 Trans-scale Quantum Science Institute, The University of Tokyo, 7-3-1, Hongo, Bunkyo-ku, Tokyo, 113-0033, Japan
4 Institute for Theoretical Physics, Center for Extreme Matter and Emergent Phenomena, Utrecht University, Princetonplein 5, 3584 CE Utrecht, The Netherlands
katsura@phys.s.u-tokyo.ac.jp, d.schuricht@uu.nl

March 28, 2025

## Abstract

We consider the valence bond solid (VBS) state built of singlet pairs of fundamental representations and projected onto adjoint representations of the exceptional Lie group $G_2$. The two-point correlation function in the VBS state is non-vanishing only for nearest neighbours, but possesses finite string order. We construct a parent Hamiltonian for the VBS state, which constitutes the $G_2$ analog of the famous AKLT chain.

**In memory of Ian Affleck.**

# 1 Introduction

In 1987 Ian Affleck together with Tom Kennedy, Elliot Lieb and Hal Tasaki introduced [1,2] a class of exactly solvable Hamiltonians. These Hamiltonians possess unique ground states constructed out of valence bonds and were thus named valence bond solids (VBSs). It was shown that there exists an energy gap and that the VBS states have short-range correlations and unbroken symmetry. Its simplest variant, now generally known as the Affleck–Kennedy–Lieb–Tasaki (AKLT) chain, corresponds to a specific point in the phase diagram [3,4] of the bilinear-biquadratic spin-1 isotropic quantum chain. In fact, the AKLT chain is located in the phase of the isotropic spin-1 Heisenberg chain (also known as the Haldane chain). Thus the physics of the AKLT chain provided a very intuitive framework for the appearance of the 'Haldane gap' [5–7], which was originally predicted using sophisticated field-theoretical arguments. More generally, VBS states helped to understand the fundamental distinction between integer and half-integer quantum spin chains [8,9], provided examples for quantum number fractionalisation and edge modes [10–12], and, in their matrix product form, constitute a natural framework for numerical approaches like the density matrix renormalisation group method [13,14]. The very intuitive construction of the AKLT chain initiated numerous generalisations, including to higher spins and higher-dimensional models (see for example [15]), but also extensions of the underlying spin symmetry to $q$-deformed [16–21] and supersymmetric [22,23] models as well as higher-dimensional Lie groups like SU($n$) [24–30], SO($n$) [31,32] and SP($n$) [33]. Here we will extend this list with $G_2$, the simplest exceptional Lie group.

In this brief note we consider an $N$-site spin chain, where the local degrees of freedom at each lattice site transform under the 14-dimensional, adjoint representation of the exceptional Lie group $G_2$. We construct a valence bond solid (VBS) state and calculate static correlation functions as well as a string order parameter. Furthermore, we derive a parent Hamiltonian of which the VBS state is a zero-energy ground state. Our results can be regarded as the $G_2$ generalisation of the famous AKLT construction.

# 2 Valence bond solid state

We consider the exceptional Lie group $G_2$ [34,35], whose Lie algebra $g_2$ is 14-dimensional. We denote the generators by $\Lambda^a$, $a = 1, \ldots, 14$. Furthermore we use bold symbols $\mathbf{\Lambda} = (\Lambda^1, \ldots, \Lambda^{14})$ to denote the vector formed by the generators. The algebra $g_2$ has rank two, thus two of the generators can be chosen to be diagonal, with the states in a given representation being labeled by their eigenvalues, see Figures 2 and 3 for examples.

The fundamental representation of $g_2$ is denoted by its dimension $\mathbf{7}$, meaning the 14 generators $\Lambda^a$ can be represented as $7 \times 7$ matrices. A possible explicit realisation is given in Appendix A. We fix the normalisation of the generators via $\mathrm{tr}[(\Lambda^a)^\dagger \Lambda^b] = 2\delta_{ab}$. The quadratic Casimir operator takes the value $\mathbf{\Lambda}^2 = \mathbf{\Lambda} \cdot \mathbf{\Lambda} = \sum_{a=1}^{14} \Lambda^a \Lambda^a = 4\,\mathrm{id} \equiv 4$. Here and in the following we assume the generators to be hermitian.

For the construction of the physical Hilbert space we consider a chain with $N$ lattice sites and periodic boundary conditions. At each lattice site we place two fundamental representations $\mathbf{7}$, see Figure 1 for a sketch. On every lattice site we thus have the tensor product (see Appendix B for more details)

$$\mathbf{7} \otimes \mathbf{7} = \mathbf{1} \oplus \mathbf{7} \oplus \mathbf{14} \oplus \mathbf{27}, \tag{1}$$

where on the right-hand side we see its decomposition into the irreducible representations $\mathbf{1}$ (singlet), $\mathbf{7}$ (fundamental), $\mathbf{14}$ (adjoint) and $\mathbf{27}$. We recall that the appearing represen-

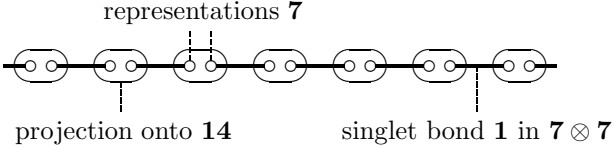

representations **7**

projection onto **14**      singlet bond **1** in **7** ⊗ **7**

Figure 1: Graphical representation of the VBS state $|\Psi_{\text{VBS}}\rangle$. Each circle stands for a fundamental representation **7**, each line joining two circles for a singlet bond, ie, the singlet **1** in the tensor product (1), and each oval for a lattice site on which we project onto the adjoint representation **14**.

tations are labeled by their dimensions. Under exchange of the factors on the left-hand side, ie, under the action of the permutation operator on **7** ⊗ **7**, the representations **1** and **27** are even, while **7** and **14** are odd. Furthermore the eigenvalues of the quadratic Casimir operator $(\mathbf{\Lambda}_1 + \mathbf{\Lambda}_2)^2$ are 0, 4, 8 and 28/3, respectively. Finally, the physical Hilbert space is obtained by projecting[1] onto the factor **14** on the right-hand side, thus obtaining a tensor product of $N$ local Hilbert spaces each transforming in the 14-dimensional, adjoint representation of $g_2$.

In this Hilbert space, the VBS state is constructed as follows: We form a singlet between one of the fundamental representations **7** on lattice site $j$ with one on the neighbouring site $j-1$, while we form another singlet with the second representation **7** on lattice site $j$ with one on the neighbouring site $j+1$. We stress that the formation of these singlets is imposed in addition to the already implemented projection onto the adjoint representation **14** at each lattice site. If we further impose periodic boundary conditions this yields a unique[2] VBS state $|\Psi_{\text{VBS}}\rangle$, which is translationally invariant and can be represented graphically as shown in Figure 1. The entanglement entropy of the VBS state can be straightforwardly obtained using its Schmidt decomposition [38]. The result in the thermodynamic limit is $S_{\text{EE}} = 2 \ln 7$, where the factor 2 stems from the fact that for the system with periondic boundary conditions two singlet bonds have to be cut, each contributing $\ln 7$.

As we show in Appendix C one can derive an explicit matrix product representation for the VBS state, which takes the form

$$|\Psi_{\text{VBS}}\rangle = \sum_{a_1, a_2, \ldots, a_N = 1}^{14} \text{tr}_{\mathbf{7}}[M^{a_1} M^{a_2} \cdots M^{a_N}] \, |\psi_{a_1}\rangle_1 |\psi_{a_2}\rangle_2 \cdots |\psi_{a_N}\rangle_N \,, \tag{2}$$

where $M^a = \Lambda^a / \sqrt{2}$ with $\Lambda^a$, $a = 1, \ldots, 14$, defined in (17) and the trace is taken over the fundamental representation **7**. Here we introduced the local states $|\psi_a\rangle_j$, $a = 1, ..., 14$, forming an orthonormal basis of the adjoint representation **14** at lattice site $j$. Explicitly,

---

[1] We note that since the representation **14** is odd under the exchange of the factors in **7**⊗**7**, the projection cannot be implemented using Schwinger bosons creating the particles in the fundamental representations, as can be done in the original AKLT chain [36, 37].

[2] In the case of open boundary conditions on the lattice sites 1 and $N$ one unpaired fundamental representation **7** would remain. Thus the construction would yield $7 \cdot 7 = 49$ VBS states.

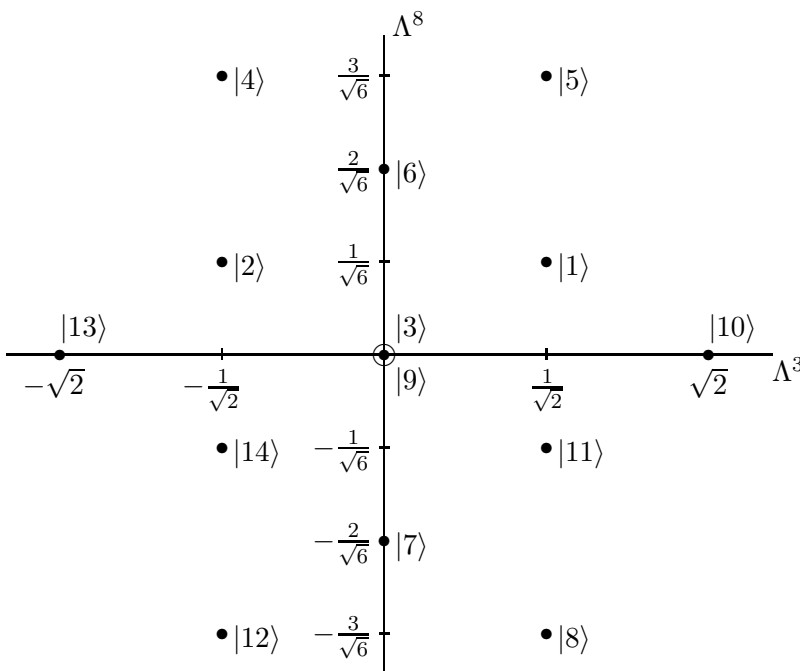

Figure 2: Weight diagram of the adjoint representation **14**. The state with $\Lambda^3 = \Lambda^8 = 0$ is doubly degenerate. An explicit expansion of the basis states in terms of the basis states of the underlying fundamental representations in the tensor product (1) is provided in Appendix C. The states appearing in the matrix product state (2) are given in (3).

these basis states are given by

$$
\begin{aligned}
|\psi_1\rangle &= \frac{|10\rangle + |13\rangle}{\sqrt{2}}, & |\psi_2\rangle &= -\mathrm{i}\frac{|10\rangle - |13\rangle}{\sqrt{2}}, & |\psi_3\rangle &= -\frac{|3\rangle + \sqrt{3}\,|9\rangle}{2}, \\
|\psi_4\rangle &= \frac{|5\rangle + |12\rangle}{\sqrt{2}}, & |\psi_5\rangle &= -\mathrm{i}\frac{|5\rangle - |12\rangle}{\sqrt{2}}, & |\psi_6\rangle &= \frac{|4\rangle + |8\rangle}{\sqrt{2}}, \\
|\psi_7\rangle &= -\mathrm{i}\frac{|4\rangle - |8\rangle}{\sqrt{2}}, & |\psi_8\rangle &= \frac{-\sqrt{3}\,|3\rangle + |9\rangle}{2}, & |\psi_9\rangle &= \mathrm{i}\frac{|6\rangle + \mathrm{i}\,|7\rangle}{\sqrt{2}}, \\
|\psi_{10}\rangle &= \frac{|6\rangle - \mathrm{i}\,|7\rangle}{\sqrt{2}}, & |\psi_{11}\rangle &= \mathrm{i}\frac{|2\rangle - \mathrm{i}\,|11\rangle}{\sqrt{2}}, & |\psi_{12}\rangle &= -\frac{|2\rangle + \mathrm{i}\,|11\rangle}{\sqrt{2}}, \\
|\psi_{13}\rangle &= -\mathrm{i}\frac{|1\rangle + \mathrm{i}\,|14\rangle}{\sqrt{2}}, & |\psi_{14}\rangle &= \frac{|1\rangle - \mathrm{i}\,|14\rangle}{\sqrt{2}},
\end{aligned}
\tag{3}
$$

where $\{|a\rangle : a = 1, \ldots, 14\}$ is another basis of the adjoint representation that diagonalises the elements of the Cartan subalgebra, ie, $\Lambda^3$ and $\Lambda^8$; we show the corresponding weight diagram in Figure 2. In Appendix C we provide an explicit basis in terms of the basis states of the underlying fundamental representations in the tensor product (1). The representation (2) will be the starting point for the calculation of the static correlation functions and string order parameters below.

## 3  Construction of a parent Hamiltonian

Having constructed the VBS state on the $N$-site chain, we now proceed in deriving a parent Hamiltonian. Specifically we construct a Hamiltonian $H_{\mathrm{AKLT}}$ which possesses the VBS state $|\Psi_{\mathrm{VBS}}\rangle$ as its zero-energy ground state.

We start by considering two neighbouring sites of the chain. We denote the local generators by $\Lambda_j^a$, $a = 1, \ldots, 14$, where the subindex indicates that they act non-trivially at site $j$ only[3]. These local generators act in the adjoint representation $\mathbf{14}$, thus on the two neighbouring sites we have to consider the tensor product

$$\mathbf{14} \otimes \mathbf{14} = \mathbf{1} \oplus \mathbf{14} \oplus \mathbf{27} \oplus \mathbf{77} \oplus \mathbf{77'}. \tag{4}$$

We note that there exist two non-isomorphic 77-dimensional representations in the decomposition. The eigenvalues of the quadratic Casimir operator $(\mathbf{\Lambda}_j + \mathbf{\Lambda}_{j+1})^2$ on the representations on the right-hand side of (4) are 0, 8, 28/3, 16 and 20, respectively. The representations $\mathbf{1}$, $\mathbf{27}$ and $\mathbf{77'}$ are even under exchange of the factors on the left-hand side, the representations $\mathbf{14}$ and $\mathbf{77}$ are odd.

The parent Hamiltonian for the VBS state is constructed by noting that on each two neighbouring sites in the VBS state we find one singlet and two uncoupled fundamental representations. Hence, the total $G_2$ contents on two neighbouring sites is given by the tensor product (1). If we construct an operator which annihilates (1) but is strictly positive on the complement of (1) in (4), we will obtain the VBS state as zero-energy ground state. Such an operator is conveniently implemented using the quadratic Casimir operator $(\mathbf{\Lambda}_j + \mathbf{\Lambda}_{j+1})^2$ on the two sites by

$$H_{j,j+1} = \frac{1}{1152}(\mathbf{\Lambda}_j + \mathbf{\Lambda}_{j+1})^2 \Big[ (\mathbf{\Lambda}_j + \mathbf{\Lambda}_{j+1})^2 - 8 \Big] \Big[ (\mathbf{\Lambda}_j + \mathbf{\Lambda}_{j+1})^2 - \frac{28}{3} \Big], \tag{5}$$

where the subindices indicate that the operator is acting locally on sites $j$ and $j + 1$. We note that the operators $H_{j,j+1}$ are not projectors[4] as they take the different values 20/27 and 20/9 on the representations $\mathbf{77}$ and $\mathbf{77'}$. The prefactor in (5) is chosen for convenience.

As the operators (5) annihilate the VBS state (2), a translationally invariant parent Hamiltonian $H_{\text{AKLT}}$ is obtained by simply summing them up for each bond on the chain. Furthermore using $\mathbf{\Lambda}_j^2 = 8$ and thus $(\mathbf{\Lambda}_j + \mathbf{\Lambda}_{j+1})^2 = 16 + 2\mathbf{\Lambda}_j \cdot \mathbf{\Lambda}_{j+1}$ we find[5]

$$H_{\text{AKLT}} = \sum_{j=1}^{N} \Big[ \frac{1}{2} \mathbf{\Lambda}_j \cdot \mathbf{\Lambda}_{j+1} + \frac{23}{216} \big( \mathbf{\Lambda}_j \cdot \mathbf{\Lambda}_{j+1} \big)^2 + \frac{1}{144} \big( \mathbf{\Lambda}_j \cdot \mathbf{\Lambda}_{j+1} \big)^3 + \frac{20}{27} \Big]. \tag{6}$$

By construction we have $H_{\text{AKLT}} |\Psi_{\text{VBS}}\rangle = 0$, and as all terms in $H_{\text{AKLT}}$ are non-negative we conclude that $|\Psi_{\text{VBS}}\rangle$ is indeed a zero-energy ground state. We stress that the generators appearing in the Hamiltonian act in the 14-dimensional, adjoint representation of $G_2$. The original AKLT chain is formulated in terms of spin-1 operators, which form the adjoint representation of SU(2). In this sense the model (6) can be regarded as the $G_2$ analog of the AKLT chain.

# 4 Static correlation functions

The explicit matrix product form (2) of the VBS state allows us to exactly calculate the static correlation functions. We follow the procedure presented for the original AKLT chain in Reference [15]. First we introduce a matrix $\tilde{M}^a$ obtained by conjugation of $M^a$,

---

[3]More precisely, they are defined as $\Lambda_j^a = \text{id}^{\otimes j-1} \otimes \Lambda^a \otimes \text{id}^{\otimes N-j}$, where id is the identity matrix of dimension 14.

[4]In fact, using the projectors onto $\mathbf{77}$ and $\mathbf{77'}$ one can construct a whole family of parent Hamiltonians. We present some details in Appendix D.

[5]We note that the Hamiltonian is manifestly interacting, in line with a general argument that $G_2$ invariance is incompatible with non-interacting systems [39].

$\tilde{M}^a_{\sigma,\sigma'} = (M^a_{\sigma,\sigma'})^*$, ie, without taking the transpose. Using this we define a $49 \times 49$ transfer matrix

$$R_{\mu,\nu} = R_{(\sigma,\tau),(\sigma',\tau')} = \sum_{a=1}^{14} \tilde{M}^a_{\sigma,\sigma'} M^a_{\tau,\tau'}, \tag{7}$$

where the indices $\mu, \nu = 1, 2, \ldots, 49$ correspond to the double indices $(1,1), (1,2), \ldots, (7,7)$. The transfer matrix $R$ is introduced for each lattice site, but given the translational invariance we suppress the index $j$. Then the squared norm of the VBS state is given by

$$\langle \Psi_{\text{VBS}} | \Psi_{\text{VBS}} \rangle = \text{tr}(R^N) = 2^N + 7 + 27 \left( -\frac{1}{3} \right)^N, \tag{8}$$

where we used that the eigenvalues of $R$ are given by 2 (unique), 1 (7-fold), $-1/3$ (27-fold) and 0 (14-fold). Note that the eigenvector corresponding to the leading eigenvalue 2 is given by $|\phi_0\rangle = \frac{1}{\sqrt{7}} \sum_{\sigma=1}^{7} |\sigma\rangle_{\mathbf{7}} |\sigma\rangle_{\mathbf{7}}$, which is a maximally entangled state[6] between the two fundamental representations.

Next we introduce a similar transfer matrix including the local action of a $G_2$ generator, namely

$$T^a_{\mu,\nu} = T^a_{(\sigma,\tau),(\sigma',\tau')} = \sum_{b,c=1}^{14} \tilde{M}^b_{\sigma,\sigma'} \langle \psi_b | \Lambda^a | \psi_c \rangle M^c_{\tau,\tau'}, \tag{9}$$

where the generator $\Lambda^a$ acts on the adjoint representation $\mathbf{14}$. Using this we obtain

$$\langle \Psi_{\text{VBS}} | \Lambda^a_1 \Lambda^b_j | \Psi_{\text{VBS}} \rangle = \text{tr}(T^a R^{j-2} T^b R^{N-j}). \tag{10}$$

Note that we have again suppressed the site indices, keeping the necessary information by the number of matrices $R$ between $T^a$ and $T^b$. The trace can be evaluated in the eigenbasis of $R$. As can be checked[7] using the explicit matrices given in Appendix A, we have $R T^b |\phi_0\rangle = 0$, thus leading to

$$\langle \Lambda^a_1 \Lambda^b_{j \geq 3} \rangle \equiv \frac{\langle \Psi_{\text{VBS}} | \Lambda^a_1 \Lambda^b_{j \geq 3} | \Psi_{\text{VBS}} \rangle}{\langle \Psi_{\text{VBS}} | \Psi_{\text{VBS}} \rangle} \sim 2^{-N} \to 0 \quad \text{for} \quad N \to \infty. \tag{11}$$

For nearest neighbours, however, the result remains finite in the thermodynamic limit and is given by

$$\langle \Lambda^a_j \Lambda^b_{j+1} \rangle \equiv \frac{\langle \Psi_{\text{VBS}} | \Lambda^a_j \Lambda^b_{j+1} | \Psi_{\text{VBS}} \rangle}{\langle \Psi_{\text{VBS}} | \Psi_{\text{VBS}} \rangle} = - \left( \frac{2}{7} + \frac{2}{3} 2^{-N} + \ldots \right) \delta_{ab}. \tag{12}$$

Thus we conclude that the static correlations in the VBS state are nearest neighbour only, in contrast to the exponentially decaying correlations in the original AKLT chain [1, 2] or various of its generalisations [17, 27, 29, 31, 33].

## 5 String order

While the correlation functions in the AKLT chain are short-ranged, hidden antiferromagnetic order exist. This can be seen by considering a non-local string operator [40, 41]

---

[6]Here we used the following property of the maximally entangled state. Let $\text{id}_{\mathbf{7}}$ be the $7 \times 7$ identity matrix. For any $7 \times 7$ matrix $A$, we have $(A \otimes \text{id}_{\mathbf{7}}) |\phi_0\rangle = (\text{id}_{\mathbf{7}} \otimes A^{\text{T}}) |\phi_0\rangle$. Noting that $R = \sum_{a=1}^{14} \tilde{M}^a \otimes M^a$, we find $R |\phi_0\rangle = \frac{1}{2} (\text{id}_{\mathbf{7}} \otimes \mathbf{\Lambda}^2) |\phi_0\rangle = 2 |\phi_0\rangle$, where we used the fact that $\mathbf{\Lambda}^2 |\sigma\rangle_{\mathbf{7}} = 4 |\sigma\rangle_{\mathbf{7}}$ for all $\sigma = 1, \ldots, 7$.

[7]More specifically we need the following properties: (i) $\langle \psi_b | \Lambda^a | \psi_c \rangle = - \langle \psi_c | \Lambda^a | \psi_b \rangle$, $a = 1, \ldots, 14$, for the generators of $g_2$ in $\mathbf{14}$ and (ii) $\sum_{b=1}^{14} \Lambda^b \Lambda^a \Lambda^b = 0$, $a = 1, \ldots, 14$, for the generators of $g_2$ in $\mathbf{7}$.

which possesses a finite expectation value. Here, the matrix product form (2) allows the calculation of a similar string order. Specifically we define the string operators

$$O_{ij}^{ab}(\alpha, \beta) = -\Lambda_i^a \exp\left(i\pi\sqrt{2}\alpha \sum_{k=i+1}^{j-1} \Lambda_k^3 + i\pi\sqrt{6}\beta \sum_{k=i+1}^{j-1} \Lambda_k^8\right) \Lambda_j^b,$$

(13)

where the string between the sites $i$ and $j$ is over the diagonal generators, which are $\Lambda^3$ and $\Lambda^8$ in our convention, see (28). Introducing a transfer matrix representing the exponential at site $k$ via

$$T(\alpha, \beta)_{\mu,\nu} = T(\alpha, \beta)_{(\sigma,\tau),(\sigma'\tau')} = \sum_{c,d=1}^{14} \tilde{M}_{\sigma,\sigma'}^c \langle\psi_c| \exp\left(i\pi\sqrt{2}\alpha\Lambda_k^3 + i\pi\sqrt{6}\beta\Lambda_k^8\right) |\psi_d\rangle M_{\tau,\tau'}^d$$

(14)

we obtain

$$\langle\Psi_{\text{VBS}}| O_{1j}^{ab}(\alpha, \beta) |\Psi_{\text{VBS}}\rangle = -\text{tr}(T^a T(\alpha, \beta)^{j-2} T^b R^{N-j}).$$

(15)

A finite result in the thermodynamic limit requires the non-vanishing matrix element of $T^a T(\alpha, \beta)^{j-2} T^b$ multiplying the $2^{N-j}$ coming from $R^{N-j}$. We have found that as $N \to \infty$

$$\langle O_{1j}^{33}(\tfrac{1}{2}, 0)\rangle \equiv \frac{\langle\Psi_{\text{VBS}}| O_{1j}^{33}(\tfrac{1}{2}, 0) |\Psi_{\text{VBS}}\rangle}{\langle\Psi_{\text{VBS}} |\Psi_{\text{VBS}}\rangle} = \frac{8}{49}\left(1 + 27\frac{(-1)^j}{6^j}\right) \xrightarrow{j\to\infty} \frac{8}{49},$$

(16)

while for example $\langle O_{1j}^{33}(\tfrac{1}{2}, \tfrac{1}{2})\rangle \sim 2^{-j}$. Thus we observe finite string order (16). It is unclear whether there exists a non-local Kennedy–Tasaki transformation [42, 43] that maps the VBS state to a classical product state. We note that on general grounds a family of models including $H_{\text{AKLT}}$ as a special case does not allow for symmetry protected topological phases with respect to $G_2$ [44]. Thus we expect that the VBS state (2) can be smoothly connected to a trivial state, in analogy to the situation for the spin-2 AKLT state [12].

# 6 Conclusion

We considered a spin chain where the local Hilbert spaces transform under the adjoint representation $\mathbf{14}$ of $G_2$, constructed a VBS state in this chain as well as a parent Hamiltonian for which the VBS state is a zero-energy ground state. Furthermore we showed that in the thermodynamic limit the static correlation functions are non-vanishing for nearest neighbours only, but that a non-local string order exists. In this sense our results can be seen as a $G_2$ analog of the famous AKLT chain.

# Acknowledgements

We would like to thank Philippe Lecheminant and Thomas Quella for useful comments on the manuscript. DS thanks the organisers of the 2025 Banff workshop "Exact Solutions in Quantum Information: Entanglement, Topology, and Quantum Circuits", where this work has been initiated. HK was supported by JSPS KAKENHI Grants No. JP23K25783, No. JP23K25790, and MEXT KAKENHI Grant-in-Aid for Transformative Research Areas A "Extreme Universe" (KAKENHI Grant No. JP21H05191).

# A   Explicit representation matrices

An explicit representation for the 14 generators $\Lambda^a$, $a = 1, \ldots, 14$, in the fundamental representation **7** can be chosen as [45]

$$
\sum_{a=1}^{14} f_a \Lambda^a = \frac{1}{\sqrt{2}} \begin{pmatrix}
0 & 0 & 0 & 0 & 0 & 0 & 0 \\
0 & f_3 & f_1 - \mathrm{i}f_2 & f_4 - \mathrm{i}f_5 & 0 & 0 & 0 \\
0 & f_1 + \mathrm{i}f_2 & -f_3 & f_6 - \mathrm{i}f_7 & 0 & 0 & 0 \\
0 & f_4 + \mathrm{i}f_5 & f_6 + \mathrm{i}f_7 & 0 & 0 & 0 & 0 \\
0 & 0 & 0 & 0 & -f_3 & -f_1 - \mathrm{i}f_2 & -f_4 - \mathrm{i}f_5 \\
0 & 0 & 0 & 0 & -f_1 + \mathrm{i}f_2 & f_3 & -f_6 - \mathrm{i}f_7 \\
0 & 0 & 0 & 0 & -f_4 + \mathrm{i}f_5 & -f_6 + \mathrm{i}f_7 & 0
\end{pmatrix}
$$

$$
+ \frac{1}{\sqrt{3}} \begin{pmatrix}
0 & f_{13} - \mathrm{i}f_{14} & f_{11} - \mathrm{i}f_{12} & \mathrm{i}f_9 + f_{10} & -\mathrm{i}f_{13} + f_{14} & -\mathrm{i}f_{11} + f_{12} & -f_9 - \mathrm{i}f_{10} \\
f_{13} + \mathrm{i}f_{14} & \frac{f_8}{\sqrt{2}} & 0 & 0 & 0 & \frac{f_9 - \mathrm{i}f_{10}}{\sqrt{2}} & \frac{\mathrm{i}f_{11} + f_{12}}{\sqrt{2}} \\
f_{11} + \mathrm{i}f_{12} & 0 & \frac{f_8}{\sqrt{2}} & 0 & -\frac{f_9 - \mathrm{i}f_{10}}{\sqrt{2}} & 0 & -\frac{\mathrm{i}f_{13} + f_{14}}{\sqrt{2}} \\
-\mathrm{i}f_9 + f_{10} & 0 & 0 & -\sqrt{2}f_8 & -\frac{\mathrm{i}f_{11} + f_{12}}{\sqrt{2}} & \frac{\mathrm{i}f_{13} + f_{14}}{\sqrt{2}} & 0 \\
\mathrm{i}f_{13} + f_{14} & 0 & -\frac{f_9 + \mathrm{i}f_{10}}{\sqrt{2}} & \frac{\mathrm{i}f_{11} - f_{12}}{\sqrt{2}} & -\frac{f_8}{\sqrt{2}} & 0 & 0 \\
\mathrm{i}f_{11} + f_{12} & \frac{f_9 + \mathrm{i}f_{10}}{\sqrt{2}} & 0 & -\frac{\mathrm{i}f_{13} - f_{14}}{\sqrt{2}} & 0 & -\frac{f_8}{\sqrt{2}} & 0 \\
-f_9 + \mathrm{i}f_{10} & -\frac{\mathrm{i}f_{11} - f_{12}}{\sqrt{2}} & \frac{\mathrm{i}f_{13} - f_{14}}{\sqrt{2}} & 0 & 0 & 0 & \sqrt{2}f_8
\end{pmatrix} . \tag{17}
$$

The so defined generators $\Lambda^a$ are hermitian and satisfy

$$
\mathrm{tr}[(\Lambda^a)^\dagger \Lambda^b] = 2\delta_{ab}, \tag{18}
$$

$$
\left[\Lambda^a, \Lambda^b\right] = \sum_{c=1}^{14} f^{abc} \Lambda^c \qquad \text{with} \qquad f^{abc} = \frac{1}{2}\mathrm{tr}\left(\left[\Lambda^a, \Lambda^b\right] \Lambda^c\right), \tag{19}
$$

$$
\mathbf{\Lambda}^2 = \sum_{a=1}^{14} \Lambda^a \Lambda^a = 4. \tag{20}
$$

The last equation states that the eigenvalue of the quadratic Casimir operator $\mathbf{\Lambda}^2$ is 4 in the fundamental representation. We note that the basis defined above exhibits the appearance of the Gell-Man matrices, showing that su(3) appears as a sub-algebra [34,45]. The weight diagram of the fundamental representation is shown in Figure 3.

The adjoint representation is 14-dimensional. Corresponding generators can be defined via $(\Lambda^a)_{bc} = f^{acb}$, the quadratic Casimir operator takes the value $\mathbf{\Lambda}^2 = 8$. Alternatively one can define the generators of **14** from the projection in the tensor product $\mathbf{7} \otimes \mathbf{7}$, see below. The weight diagram is shown in Figure 2.

# B   On the tensor product $\mathbf{7} \otimes \mathbf{7}$

The projectors onto the irreducible representations in the decomposition of the tensor product (1) are explicitly given by

$$
P_{\mathbf{1}} = -\frac{3}{896}\left[(\mathbf{\Lambda}_1 + \mathbf{\Lambda}_2)^2 - 4\right]\left[(\mathbf{\Lambda}_1 + \mathbf{\Lambda}_2)^2 - 8\right]\left[(\mathbf{\Lambda}_1 + \mathbf{\Lambda}_2)^2 - \frac{28}{3}\right], \tag{21}
$$

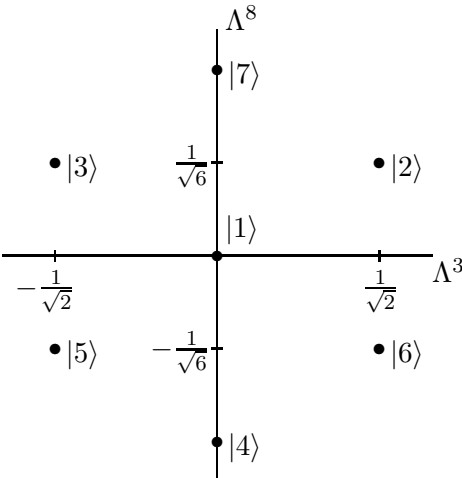

Figure 3: Weight diagram of the fundamental representation **7**. The dots represent eigenstates $|\sigma\rangle$, $\sigma = 1, \ldots, 7$, of the two diagonal generators $\Lambda^3$ and $\Lambda^8$, as fixed by the explicit matrix representation (17).

$$P_{\mathbf{7}} = \frac{3}{256}(\mathbf{\Lambda}_1 + \mathbf{\Lambda}_2)^2 \Big[(\mathbf{\Lambda}_1 + \mathbf{\Lambda}_2)^2 - 8\Big]\Big[(\mathbf{\Lambda}_1 + \mathbf{\Lambda}_2)^2 - \frac{28}{3}\Big], \tag{22}$$

$$P_{\mathbf{14}} = -\frac{3}{128}(\mathbf{\Lambda}_1 + \mathbf{\Lambda}_2)^2 \Big[(\mathbf{\Lambda}_1 + \mathbf{\Lambda}_2)^2 - 4\Big]\Big[(\mathbf{\Lambda}_1 + \mathbf{\Lambda}_2)^2 - \frac{28}{3}\Big], \tag{23}$$

$$P_{\mathbf{27}} = \frac{27}{1792}(\mathbf{\Lambda}_1 + \mathbf{\Lambda}_2)^2 \Big[(\mathbf{\Lambda}_1 + \mathbf{\Lambda}_2)^2 - 4\Big]\Big[(\mathbf{\Lambda}_1 + \mathbf{\Lambda}_2)^2 - 8\Big]. \tag{24}$$

Here $\mathbf{\Lambda}_{1,2}$ denote the generators in the first and second fundamental representation **7** respectively. In the following we use the explicit matrices given in Appendix A, ie, we fix the basis $\{|\sigma\rangle_{1,2} : \sigma = 1, \ldots, 7\}$. Then the unique singlet state can be written as

$$
\begin{aligned}
| \hspace{-0.5em}\circ\!\!-\!\!\circ\hspace{-0.5em} \rangle &= \frac{1}{\sqrt{7}}\Big(\mathrm{i}\,|1\rangle_1\,|1\rangle_2 + |2\rangle_1\,|5\rangle_2 + |3\rangle_1\,|6\rangle_2 + |4\rangle_1\,|7\rangle_2 + |5\rangle_1\,|2\rangle_2 + |6\rangle_1\,|3\rangle_2 + |7\rangle_1\,|4\rangle_2\Big) \\
&= \frac{1}{\sqrt{7}}\Big(\mathrm{i}\,|1\rangle_1\,,|2\rangle_1\,,|3\rangle_1\,,\ldots,|7\rangle_1\Big)\Big(|1\rangle_2\,,|5\rangle_2\,,|6\rangle_2\,,|7\rangle_2\,,|2\rangle_2\,,|3\rangle_2\,,|4\rangle_2\Big)^{\mathrm{t}} \\
&= \frac{1}{\sqrt{7}}\,\vec{v}_1\vec{w}_2^{\,\mathrm{t}}, \tag{25}
\end{aligned}
$$

where in the last line we have introduced auxiliary vectors $\vec{v}_1$ and $\vec{w}_2$ with elements in the local Hilbert spaces of $\mathbf{7}_{1,2}$. We note that the singlet state is even under exchange of the factors. Similarly, a basis of the adjoint representation can be obtained from the eigenspace of the projector $P_{\mathbf{14}}$, see Appendix C below.

## C  Matrix product representation of the VBS state

In this appendix we derive the explicit matrices $M^a$ in the matrix product representation (2). We start with a singlet formed by two fundamental representations at neighbouring lattices $j$ and $j + 1$, which according to (25) is given by $\vec{v}_j\vec{w}_{j+1}^{\,\mathrm{t}}$ and we dropped the normalisation. Thus considering two singlets on the bonds $(j - 1, j)$ and $(j, j + 1)$ we find

$$| \hspace{-0.3em}\circ\!\!-\!\!\circ \ \ \circ\!\!-\!\!\circ\hspace{-0.3em} \rangle \propto \vec{v}_{j-1}\vec{w}_j^{\,\mathrm{t}}\,\vec{v}_j\vec{w}_{j+1}^{\,\mathrm{t}} = \vec{v}_{j-1}\,\mathcal{M}_j^{\mathbf{7}\otimes\mathbf{7}}\,\vec{w}_{j+1}^{\,\mathrm{t}}, \tag{26}$$

where we introduced the matrix $\mathcal{M}_j^{\mathbf{7}\otimes\mathbf{7}} = \vec{w}_j^{\,\mathrm{t}}\,\vec{v}_j$ which takes values in the tensor product $\mathbf{7}\otimes\mathbf{7}$ at lattice site $j$. Explicitly we find

$$
\mathcal{M}_j^{\mathbf{7}\otimes\mathbf{7}} = \begin{pmatrix}
\mathrm{i}\,|1\rangle_{\mathbf{7}}\,|1\rangle_{\mathbf{7}} & |1\rangle_{\mathbf{7}}\,|2\rangle_{\mathbf{7}} & |1\rangle_{\mathbf{7}}\,|3\rangle_{\mathbf{7}} & |1\rangle_{\mathbf{7}}\,|4\rangle_{\mathbf{7}} & |1\rangle_{\mathbf{7}}\,|5\rangle_{\mathbf{7}} & |1\rangle_{\mathbf{7}}\,|6\rangle_{\mathbf{7}} & |1\rangle_{\mathbf{7}}\,|7\rangle_{\mathbf{7}} \\
\mathrm{i}\,|5\rangle_{\mathbf{7}}\,|1\rangle_{\mathbf{7}} & |5\rangle_{\mathbf{7}}\,|2\rangle_{\mathbf{7}} & |5\rangle_{\mathbf{7}}\,|3\rangle_{\mathbf{7}} & |5\rangle_{\mathbf{7}}\,|4\rangle_{\mathbf{7}} & |5\rangle_{\mathbf{7}}\,|5\rangle_{\mathbf{7}} & |5\rangle_{\mathbf{7}}\,|6\rangle_{\mathbf{7}} & |5\rangle_{\mathbf{7}}\,|7\rangle_{\mathbf{7}} \\
\mathrm{i}\,|6\rangle_{\mathbf{7}}\,|1\rangle_{\mathbf{7}} & |6\rangle_{\mathbf{7}}\,|2\rangle_{\mathbf{7}} & |6\rangle_{\mathbf{7}}\,|3\rangle_{\mathbf{7}} & |6\rangle_{\mathbf{7}}\,|4\rangle_{\mathbf{7}} & |6\rangle_{\mathbf{7}}\,|5\rangle_{\mathbf{7}} & |6\rangle_{\mathbf{7}}\,|6\rangle_{\mathbf{7}} & |6\rangle_{\mathbf{7}}\,|7\rangle_{\mathbf{7}} \\
\mathrm{i}\,|7\rangle_{\mathbf{7}}\,|1\rangle_{\mathbf{7}} & |7\rangle_{\mathbf{7}}\,|2\rangle_{\mathbf{7}} & |7\rangle_{\mathbf{7}}\,|3\rangle_{\mathbf{7}} & |7\rangle_{\mathbf{7}}\,|4\rangle_{\mathbf{7}} & |7\rangle_{\mathbf{7}}\,|5\rangle_{\mathbf{7}} & |7\rangle_{\mathbf{7}}\,|6\rangle_{\mathbf{7}} & |7\rangle_{\mathbf{7}}\,|7\rangle_{\mathbf{7}} \\
\mathrm{i}\,|2\rangle_{\mathbf{7}}\,|1\rangle_{\mathbf{7}} & |2\rangle_{\mathbf{7}}\,|2\rangle_{\mathbf{7}} & |2\rangle_{\mathbf{7}}\,|3\rangle_{\mathbf{7}} & |2\rangle_{\mathbf{7}}\,|4\rangle_{\mathbf{7}} & |2\rangle_{\mathbf{7}}\,|5\rangle_{\mathbf{7}} & |2\rangle_{\mathbf{7}}\,|6\rangle_{\mathbf{7}} & |2\rangle_{\mathbf{7}}\,|7\rangle_{\mathbf{7}} \\
\mathrm{i}\,|3\rangle_{\mathbf{7}}\,|1\rangle_{\mathbf{7}} & |3\rangle_{\mathbf{7}}\,|2\rangle_{\mathbf{7}} & |3\rangle_{\mathbf{7}}\,|3\rangle_{\mathbf{7}} & |3\rangle_{\mathbf{7}}\,|4\rangle_{\mathbf{7}} & |3\rangle_{\mathbf{7}}\,|5\rangle_{\mathbf{7}} & |3\rangle_{\mathbf{7}}\,|6\rangle_{\mathbf{7}} & |3\rangle_{\mathbf{7}}\,|7\rangle_{\mathbf{7}} \\
\mathrm{i}\,|4\rangle_{\mathbf{7}}\,|1\rangle_{\mathbf{7}} & |4\rangle_{\mathbf{7}}\,|2\rangle_{\mathbf{7}} & |4\rangle_{\mathbf{7}}\,|3\rangle_{\mathbf{7}} & |4\rangle_{\mathbf{7}}\,|4\rangle_{\mathbf{7}} & |4\rangle_{\mathbf{7}}\,|5\rangle_{\mathbf{7}} & |4\rangle_{\mathbf{7}}\,|6\rangle_{\mathbf{7}} & |4\rangle_{\mathbf{7}}\,|7\rangle_{\mathbf{7}}
\end{pmatrix}.
$$
(27)

Here the subscripts indicate that the states belong to the fundamental representations $\mathbf{7}$. The next step is now to project the states in the matrix $\mathcal{M}_j^{\mathbf{7}\otimes\mathbf{7}}$ onto the adjoint representation $\mathbf{14}$. To this end we obtain the representation space as the eigenspace of the projector $P_{\mathbf{14}}$ and within this choose the orthonormal basis (with the subscripts indicating the respective representations)

$$
\begin{aligned}
|1\rangle_{\mathbf{14}} &= -\frac{1}{\sqrt{3}}\Big(|1\rangle_{\mathbf{7}}\,|2\rangle_{\mathbf{7}} - |2\rangle_{\mathbf{7}}\,|1\rangle_{\mathbf{7}}\Big) - \frac{1}{\sqrt{6}}\Big(|6\rangle_{\mathbf{7}}\,|7\rangle_{\mathbf{7}} - |7\rangle_{\mathbf{7}}\,|6\rangle_{\mathbf{7}}\Big), \\
|2\rangle_{\mathbf{14}} &= \frac{1}{\sqrt{3}}\Big(|1\rangle_{\mathbf{7}}\,|3\rangle_{\mathbf{7}} - |3\rangle_{\mathbf{7}}\,|1\rangle_{\mathbf{7}}\Big) - \frac{1}{\sqrt{6}}\Big(|5\rangle_{\mathbf{7}}\,|7\rangle_{\mathbf{7}} - |7\rangle_{\mathbf{7}}\,|5\rangle_{\mathbf{7}}\Big), \\
|3\rangle_{\mathbf{14}} &= \frac{1}{2}\Big(|2\rangle_{\mathbf{7}}\,|5\rangle_{\mathbf{7}} - |5\rangle_{\mathbf{7}}\,|2\rangle_{\mathbf{7}}\Big) - \frac{1}{2}\Big(|4\rangle_{\mathbf{7}}\,|7\rangle_{\mathbf{7}} - |7\rangle_{\mathbf{7}}\,|4\rangle_{\mathbf{7}}\Big), \\
|4\rangle_{\mathbf{14}} &= -\frac{1}{\sqrt{2}}\Big(|3\rangle_{\mathbf{7}}\,|7\rangle_{\mathbf{7}} - |7\rangle_{\mathbf{7}}\,|3\rangle_{\mathbf{7}}\Big), \\
|5\rangle_{\mathbf{14}} &= -\frac{1}{\sqrt{2}}\Big(|2\rangle_{\mathbf{7}}\,|7\rangle_{\mathbf{7}} - |7\rangle_{\mathbf{7}}\,|2\rangle_{\mathbf{7}}\Big), \\
|6\rangle_{\mathbf{14}} &= -\frac{1}{\sqrt{3}}\Big(|1\rangle_{\mathbf{7}}\,|7\rangle_{\mathbf{7}} - |7\rangle_{\mathbf{7}}\,|1\rangle_{\mathbf{7}}\Big) + \frac{\mathrm{i}}{\sqrt{6}}\Big(|2\rangle_{\mathbf{7}}\,|3\rangle_{\mathbf{7}} - |3\rangle_{\mathbf{7}}\,|2\rangle_{\mathbf{7}}\Big), \\
|7\rangle_{\mathbf{14}} &= -\frac{1}{\sqrt{3}}\Big(|1\rangle_{\mathbf{7}}\,|4\rangle_{\mathbf{7}} - |4\rangle_{\mathbf{7}}\,|1\rangle_{\mathbf{7}}\Big) - \frac{1}{\sqrt{6}}\Big(|5\rangle_{\mathbf{7}}\,|6\rangle_{\mathbf{7}} - |6\rangle_{\mathbf{7}}\,|5\rangle_{\mathbf{7}}\Big), \\
|8\rangle_{\mathbf{14}} &= -\frac{1}{\sqrt{2}}\Big(|4\rangle_{\mathbf{7}}\,|6\rangle_{\mathbf{7}} - |6\rangle_{\mathbf{7}}\,|4\rangle_{\mathbf{7}}\Big), \\
|9\rangle_{\mathbf{14}} &= \frac{1}{2\sqrt{3}}\Big(|2\rangle_{\mathbf{7}}\,|5\rangle_{\mathbf{7}} - |5\rangle_{\mathbf{7}}\,|2\rangle_{\mathbf{7}}\Big) - \frac{1}{\sqrt{3}}\Big(|3\rangle_{\mathbf{7}}\,|6\rangle_{\mathbf{7}} - |6\rangle_{\mathbf{7}}\,|3\rangle_{\mathbf{7}}\Big) \\
&\quad + \frac{1}{2\sqrt{3}}\Big(|4\rangle_{\mathbf{7}}\,|7\rangle_{\mathbf{7}} - |7\rangle_{\mathbf{7}}\,|4\rangle_{\mathbf{7}}\Big), \\
|10\rangle_{\mathbf{14}} &= -\frac{1}{\sqrt{2}}\Big(|2\rangle_{\mathbf{7}}\,|6\rangle_{\mathbf{7}} - |6\rangle_{\mathbf{7}}\,|2\rangle_{\mathbf{7}}\Big), \\
|11\rangle_{\mathbf{14}} &= -\frac{1}{\sqrt{3}}\Big(|1\rangle_{\mathbf{7}}\,|6\rangle_{\mathbf{7}} - |6\rangle_{\mathbf{7}}\,|1\rangle_{\mathbf{7}}\Big) - \frac{\mathrm{i}}{\sqrt{6}}\Big(|2\rangle_{\mathbf{7}}\,|4\rangle_{\mathbf{7}} - |4\rangle_{\mathbf{7}}\,|2\rangle_{\mathbf{7}}\Big), \\
|12\rangle_{\mathbf{14}} &= -\frac{1}{\sqrt{2}}\Big(|4\rangle_{\mathbf{7}}\,|5\rangle_{\mathbf{7}} - |5\rangle_{\mathbf{7}}\,|4\rangle_{\mathbf{7}}\Big), \\
|13\rangle_{\mathbf{14}} &= -\frac{1}{\sqrt{2}}\Big(|3\rangle_{\mathbf{7}}\,|5\rangle_{\mathbf{7}} - |5\rangle_{\mathbf{7}}\,|3\rangle_{\mathbf{7}}\Big), \\
|14\rangle_{\mathbf{14}} &= -\frac{1}{\sqrt{3}}\Big(|1\rangle_{\mathbf{7}}\,|5\rangle_{\mathbf{7}} - |5\rangle_{\mathbf{7}}\,|1\rangle_{\mathbf{7}}\Big) + \frac{\mathrm{i}}{\sqrt{6}}\Big(|3\rangle_{\mathbf{7}}\,|4\rangle_{\mathbf{7}} - |4\rangle_{\mathbf{7}}\,|3\rangle_{\mathbf{7}}\Big).
\end{aligned}
$$

The states are shown in the weight diagram of the representation $\mathbf{14}$ in Figure 2. We note that the basis vectors are indeed odd under exchange of the factors. One can straightforwardly check that indeed $_{\mathbf{14}}\langle a|b\rangle_{\mathbf{14}} = \delta_{ab}$. The action of the generators $\mathbf{\Lambda}$ on the basis

states $|a\rangle_{\mathbf{14}}$ is obtained from their action on the respective factors. For example,

$$\Lambda^1 |10\rangle_{\mathbf{14}} = \left(\Lambda_1^1 + \Lambda_2^1\right) \left[ -\frac{1}{\sqrt{2}} \left( |2\rangle_{\mathbf{7}} |6\rangle_{\mathbf{7}} - |6\rangle_{\mathbf{7}} |2\rangle_{\mathbf{7}} \right) \right] = \frac{1}{2} |3\rangle_{\mathbf{14}} + \frac{\sqrt{3}}{2} |9\rangle_{\mathbf{14}} . \tag{28}$$

Using this the action of the generators on the matrix $\mathcal{M}_j$ required in the calculation of the static correlation functions and string order parameter can be worked out.

Finally, to obtain the explicit expressions of $M^a$ we project the matrix $\mathcal{M}_j^{\mathbf{7} \otimes \mathbf{7}}$ in (27) onto the basis $\{|a\rangle_{\mathbf{14}} : a = 1, \ldots, 14\}$ and dropping the subindex $\mathbf{14}$. As a result we have

$$\mathcal{M}_j^{\mathbf{14}} = \begin{pmatrix} 0 & -\frac{1}{\sqrt{3}}|1\rangle & \frac{1}{\sqrt{3}}|2\rangle & -\frac{1}{\sqrt{3}}|7\rangle & -\frac{1}{\sqrt{3}}|14\rangle & -\frac{1}{\sqrt{3}}|11\rangle & -\frac{1}{\sqrt{3}}|6\rangle \\ \frac{i}{\sqrt{3}}|14\rangle & -\frac{1}{2}|3\rangle - \frac{1}{2\sqrt{3}}|9\rangle & \frac{1}{\sqrt{2}}|13\rangle & \frac{1}{\sqrt{2}}|12\rangle & 0 & -\frac{1}{\sqrt{6}}|7\rangle & -\frac{1}{\sqrt{6}}|2\rangle \\ \frac{i}{\sqrt{3}}|11\rangle & \frac{1}{\sqrt{2}}|10\rangle & \frac{1}{\sqrt{3}}|9\rangle & \frac{1}{\sqrt{2}}|8\rangle & \frac{1}{\sqrt{6}}|7\rangle & 0 & -\frac{1}{\sqrt{6}}|1\rangle \\ \frac{i}{\sqrt{3}}|6\rangle & \frac{1}{\sqrt{2}}|5\rangle & \frac{1}{\sqrt{2}}|4\rangle & \frac{1}{2}|3\rangle - \frac{1}{2\sqrt{3}}|9\rangle & \frac{1}{\sqrt{6}}|2\rangle & \frac{1}{\sqrt{6}}|1\rangle & 0 \\ \frac{i}{\sqrt{3}}|1\rangle & 0 & -\frac{1}{\sqrt{6}}|6\rangle & \frac{i}{\sqrt{6}}|11\rangle & \frac{1}{2}|3\rangle + \frac{1}{2\sqrt{3}}|9\rangle & -\frac{1}{\sqrt{2}}|10\rangle & -\frac{1}{\sqrt{2}}|5\rangle \\ -\frac{i}{\sqrt{3}}|2\rangle & \frac{i}{\sqrt{6}}|6\rangle & 0 & -\frac{i}{\sqrt{6}}|14\rangle & -\frac{1}{\sqrt{2}}|13\rangle & -\frac{1}{\sqrt{3}}|9\rangle & -\frac{1}{\sqrt{2}}|4\rangle \\ \frac{i}{\sqrt{3}}|7\rangle & -\frac{i}{\sqrt{6}}|11\rangle & \frac{i}{\sqrt{6}}|14\rangle & 0 & -\frac{1}{\sqrt{2}}|12\rangle & -\frac{1}{\sqrt{2}}|8\rangle & -\frac{1}{2}|3\rangle + \frac{1}{2\sqrt{3}}|9\rangle \end{pmatrix} . \tag{29}$$

This matrix can be rewritten in terms of the basis states $|\psi_a\rangle$ in (3) as

$$\mathcal{M}_j^{\mathbf{14}} = \frac{1}{\sqrt{2}} \sum_{a=1}^{14} \check{\Lambda}^a |\psi_a\rangle_j , \tag{30}$$

where $\check{\Lambda}^a = V \Lambda^a V^\dagger$ with $V = \mathrm{diag}(-\mathrm{i}, 1, 1, 1, 1, 1, 1)$ and the index $j$ on the right-hand side was added to indicate that $|\psi_a\rangle$ is a local state at site $j$. Using this we obtain

$$|\Psi_{\mathrm{VBS}}\rangle = \mathrm{tr}_{\mathbf{7}} \left[ \mathcal{M}_1 \mathcal{M}_2 \cdots \mathcal{M}_N \right]$$

$$= \frac{1}{2^{N/2}} \sum_{a_1, a_2, \ldots, a_N = 1}^{14} \mathrm{tr}_{\mathbf{7}} \left[ \check{\Lambda}^{a_1} \check{\Lambda}^{a_2} \cdots \check{\Lambda}^{a_N} \right] |\psi_{a_1}\rangle_1 |\psi_{a_2}\rangle_2 \cdots |\psi_{a_N}\rangle_N , \tag{31}$$

where the trace is taken over the 7-dimensional auxiliary space. In the trace the matrices $V$ and $V^\dagger$ cancel out in pairs, and we are left with

$$|\Psi_{\mathrm{VBS}}\rangle = \frac{1}{2^{N/2}} \sum_{a_1, a_2, \ldots, a_N = 1}^{14} \mathrm{tr}_{\mathbf{7}} \left[ \Lambda^{a_1} \Lambda^{a_2} \cdots \Lambda^{a_N} \right] |\psi_{a_1}\rangle_1 |\psi_{a_2}\rangle_2 \cdots |\psi_{a_N}\rangle_N , \tag{32}$$

which is the desired (2).

## D  Family of parent Hamitonians

Using the projectors onto the representations $\mathbf{77}$ and $\mathbf{77'}$ in (4) one can construct a family of parent Hamiltonians for the VBS state (2). We start from the projectors

$$P_{\mathbf{77}} = -\frac{3}{10240} (\boldsymbol{\Lambda}_1 + \boldsymbol{\Lambda}_2)^2 \left[ (\boldsymbol{\Lambda}_1 + \boldsymbol{\Lambda}_2)^2 - 8 \right] \left[ (\boldsymbol{\Lambda}_1 + \boldsymbol{\Lambda}_2)^2 - \frac{28}{3} \right] \left[ (\boldsymbol{\Lambda}_1 + \boldsymbol{\Lambda}_2)^2 - 20 \right],$$

$$P_{\mathbf{77'}} = \frac{1}{10240} (\boldsymbol{\Lambda}_1 + \boldsymbol{\Lambda}_2)^2 \left[ (\boldsymbol{\Lambda}_1 + \boldsymbol{\Lambda}_2)^2 - 8 \right] \left[ (\boldsymbol{\Lambda}_1 + \boldsymbol{\Lambda}_2)^2 - \frac{28}{3} \right] \left[ (\boldsymbol{\Lambda}_1 + \boldsymbol{\Lambda}_2)^2 - 16 \right],$$

where $\boldsymbol{\Lambda}_{1,2}$ denote the generators on the respective factors in $\mathbf{14} \otimes \mathbf{14}$. Using these an operator which annihilates (1) and is strictly positive on the complement of (1) in (4) is given by

$$\tilde{H}_{j,j+1} = a\, P_{\mathbf{77}} + b\, P_{\mathbf{77'}} = \sum_{n=0}^{4} \mu_n \left( \boldsymbol{\Lambda}_j \cdot \boldsymbol{\Lambda}_{j+1} \right)^n , \tag{33}$$

where $a, b > 0$ and

$$\mu_0 = a, \ \mu_1 = \frac{7a}{40} + \frac{b}{6}, \ \mu_2 = -\frac{31a}{160} + \frac{9b}{80}, \ \mu_3 = -\frac{a}{16} + \frac{23b}{960}, \ \mu_4 = -\frac{3a}{640} + \frac{b}{640}. \quad (34)$$

The parameters $a$ and $b$ can be tuned such that one of the coefficients $\mu_n$, $n = 2, 3, 4$, vanishes. For example, choosing $b = 3a$ implies $\mu_4 = 0$ [with $a = 20/27$ giving (5)]; in this sense the Hamiltonian (6) is the simplest parent Hamiltonian for the VBS state.

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
