# Peer review of "A brief note on the G$_2$ Affleck-Kennedy-Lieb-Tasaki chain"

_SciPost Physics Core_

## Round 2 · Referee Report · Anonymous (Referee 1) · 2025-4-30

Strengths

see report

Weaknesses

see report

Report

Referee Report

The manuscript entitled “A brief note on the G2 Affleck–Kennedy–Lieb–Tasaki chain” is an excellent short paper on the valence bond solid (VBS) with G2 symmetry. The authors provide a concise and well-rounded introduction and review of the construction by Affleck–Kennedy–Lieb–Tasaki.

The fundamental representation of G2 is 7-dimensional. On each lattice site, two such representations are introduced. In the decomposition of their tensor product, the representations 1 ⊕ 7 ⊕ 14 ⊕ 27 appear, of which all except the 14-dimensional adjoint representation are projected out.

The resulting rank-3 tensor serves as the basic building block for the multi-spin state composed of 14-dimensional adjoint representations: neighbouring tensors are connected by contracting their adjacent 7-dimensional components into singlets. Locally, the product of two adjoint (14-dimensional) representations may contain 1 ⊕ 14 ⊕ 27 ⊕ 77 ⊕ 77′. However, due to the specific construction of the VBS state — starting from four 7-dimensional representations of which two are contracted to a singlet — at most 1 ⊕ 7 ⊕ 14 ⊕ 27 can appear, resulting in 1 ⊕ 14 ⊕ 27.

The parental Hamiltonian, designed to take this constructed VBS state as its ground state, is chosen to annihilate the 1 ⊕ 14 ⊕ 27 components while assigning finite positive energy to the 77 ⊕ 77′ components. The authors investigate the two-point correlation function via a transfer matrix method and find it to be non-vanishing only for nearest neighbours. However, a string order parameter can be defined, which turns out to be finite.

The paper is well written, very accessible, and the results are, to my knowledge, new. The main body is presented with minimal technical detail — the authors rely primarily on fundamental properties of tensor product decompositions and the eigenvalues of the quadratic Casimir operator. The actual computations begin with the evaluation of the correlation function and the string order parameter, with technical details deferred to the appendices.

I would, however, suggest that the authors comment on the uniqueness of the ground state (not just of the VBS state itself). Typically, one would need to prove that the excited states remain gapped, even in the thermodynamic limit, to confirm uniqueness — a technically demanding part of the original AKLT construction. It would be valuable to clarify whether the G2 symmetry affects this aspect or simplifies the situation in any way.

I recommend this manuscript for publication in SciPost.

discovered "typos":

can be regarded as the G2 generalisation of the famous AKLT construction. -> can be regarded as the G2 version/analog of the famous AKLT construction.

ie -> i.e.

parent Hamiltonian -> parental Hamiltonian

hidden antiferro-magnetic order exist. -> hidden antiferro-magnetic order exists.

subindex -> index or subscript

Recommendation

Publish (surpasses expectations and criteria for this Journal; among top 10%)

---

## Round 2 · Referee Report · Anonymous (Referee 2) · 2025-7-1

Strengths

1-This is a solid and detailed analysis of the $G_2$ AKLT state in the adjoint representation
2-The presentation is very detailed and easy to follow
3-The state appears to exhibit some non-standard features such as a vanishing two-point correlation function beyond nearest neighbors

Weaknesses

1-It is not immediately obvious how this model can be realized in a more physical context

Report

This is a very nice though rather technical paper with results that are new to my knowledge. All relevant quantities are introduced in detail, the physical and representation theoretic reasonings are correct and the results seem to make sense. As the paper extends the AKLT construction to $G_2$ it is a fantastic fit for the Ian Affleck memorial volume. I agree with the other referee that a comment on the uniqueness of the ground state (and the existence of a gap) would be valuable.

I checked various of the technical statements in detail and could not find a mistake. Still, I would recommend to comment more clearly on the role of the two different bases $|\psi_a\rangle$ and $|a\rangle$. Specifically, it should be mentioned that the $\Lambda^a$ in the adjoint representation as defined at the end of Appendix A act on the $|\psi_a\rangle$. This is sort of clear given that $\Lambda^3$ and $\Lambda^8$ are not diagonal in the adjoint representation (if one works out the matrices) [which is the reason to introduce the $|a\rangle$] and also from footnote 7 where the matrix elements appear and are (implicitly) identified with the structure constants to get the antisymmetry property. However, I started a computer implementation based on the information in the appendices and got quite confused initially.

Requested changes

There are a few typos and I leave it to the discretion of the authors to address these
1-"periondic" above Eq (2)
2-In footnote 7 there are two incarnations of $g_2$ instead of $G_2$
3-In contrast to the other referee I would suggest to stick to "parent Hamiltonian"
4-Please clarify the role of the two different bases in the adjoint representation (see above)

Recommendation

Publish (easily meets expectations and criteria for this Journal; among top 50%)

---

## Editorial Decision

resubmitted